# Comparing Atrial-Fibrillation Validated Rapid Scoring Systems in the Long-Term Mortality Prediction in Patients Referred for Elective Coronary Angiography: A Subanalysis of the Białystok Coronary Project

**DOI:** 10.3390/ijerph191610419

**Published:** 2022-08-21

**Authors:** Ewelina Rogalska, Anna Kurasz, Łukasz Kuźma, Hanna Bachórzewska-Gajewska, Sławomir Dobrzycki, Marek Koziński, Bożena Sobkowicz, Anna Tomaszuk-Kazberuk

**Affiliations:** 1Department of Cardiology, Medical University of Białystok, 24A Skłodowskiej-Curie, 15-276 Białystok, Poland; 2Department of Invasive Cardiology, Medical University of Białystok, 24A Skłodowskiej-Curie, 15-276 Białystok, Poland; 3Department of Cardiology and Internal Medicine, Medical University of Gdańsk, 9b Powstania Styczniowego, 81-519 Gdynia, Poland

**Keywords:** atrial fibrillation, mortality, CHA_2_DS_2_-VASc, HAS–BLED, 2MACE

## Abstract

Rapid scoring systems validated in patients with atrial fibrillation (AF) may be useful beyond their original purpose. Our aim was to assess the utility of CHA_2_DS_2_-VASc, HAS–BLED, and 2MACE scores in predicting long-term mortality in the population of the Białystok Coronary Project, including AF patients. The initial study population consisted of 7409 consecutive patients admitted for elective coronary angiography between 2007 and 2016. The study endpoint was all-cause mortality, which occurred in 1244 (16.8%) patients during the follow-up, ranging from 1283 to 3059 days (median 2029 days). We noticed substantially increased all-cause mortality in patients with higher values of all compared scores. The accuracy of the scores in predicting all-cause mortality was also assessed using the receiver operator characteristic (ROC) curves. The greatest predictive value for mortality was recorded for the CHA_2_DS_2_-VASc score in the overall study population (area under curve [AUC] = 0.665; 95% confidence interval [95%CI] 0.645–0.681). We observed that the 2MACE score (AUC = 0.656; 95%CI 0.619–0.681), but not the HAS–BLED score, had similar predictive value to the CHA_2_DS_2_-VASc score for all-cause mortality in the overall study population. In AF patients, all scores did not differ in all-cause mortality prediction. Additionally, we found that study participants with CHA_2_DS_2_-VASc score ≥3 vs. <3 had a 3-fold increased risk of long-term all-cause mortality (odds ratio 3.05; 95%CI 2.6–3.6). Our study indicates that clinical scores initially validated in AF patients may be useful for predicting mortality in a broader population (e.g., in patients referred for elective coronary angiography). According to our findings, all compared scores have a moderate predictive value. However, in our study, the CHA_2_DS_2_-VASc and 2MACE scores outperformed the HAS–BLED score in terms of the long-term all-cause mortality prediction.

## 1. Introduction

The Bialystok Coronary Project (AF-CAD study) is an observational study of patients referred for elective coronary angiography, focused on the co-occurrence of coronary artery disease (CAD) and atrial fibrillation (AF). In the first publication, we showed that AF is associated with a lack of significant coronary lesions in coronary angiography, which reflects the difficulties in qualifying these patients for invasive diagnostics of chronic coronary syndromes [1]. The study cohort was further followed up and analyzed in order to distinguish the predictors of mortality [2].

Nowadays, especially in the light of the COVID-19 pandemic, there is a growing problem of limited physician availability, and telemedicine is playing an increasing role [3]. This enhances the potential and utility of predictive scales, which are designed to enable physicians to assess a patient’s prognosis. This allows for the identification of vulnerable groups or those in immediate need of hospitalization based on simple, additive scoring systems when face-to-face contact is either difficult or even impossible. The most widely known scales that can be used to assess general cardiovascular risk are the Framingham Risk Score and the SCORE Risk Charts [4,5]. These scales have been used since the second half of the 20th century, and since then, their local versions have been created, e.g., POL-SCORE, which allows better adjustment of factors included in the scale to the examined population [6]. In the case of AF patients, the following scores have been developed to assess the risk of: (i) thromboembolic complications (CHA_2_DS_2_-VASc), (ii) major bleeding (HAS-BLED), and (iii) composite outcomes (2MACE) [7,8].

An aging population, undeniable medical advances, and increased life expectancy make it increasingly challenging to identify new risk factors. Things that once posed a direct risk of death, such as infections, are now giving way to new risk factors that were once neglected, such as air pollution and climate change [9,10]. Therefore, a continuous validation of the existing scales and consideration of new potential risk factors beyond those strictly related to the patient’s condition are essential.

Given the simplicity of using these AF scales, we aimed to test their usefulness beyond their original purpose, namely in the prediction of mortality. Since we wanted to achieve results referring to a broader population, we tested the utility of these scales not only in the AF subgroup but also in the overall study population. Therefore, our aim was to compare rapid scoring systems in the long-term all-cause mortality prediction in the cohort of the Białystok Coronary Project (AF-CAD study).

## 2. Materials and Methods

### 2.1. Study Design

The Białystok Coronary Project (AF-CAD study) is a retrospective cohort study of consecutive patients with confirmed or suspected CAD conducted in the Department of Invasive Cardiology of the Medical University of Bialystok, Poland. Study details and outcomes are presented in a previous publication [1].

Briefly, study participants were recruited between 2007 and 2016. We based diagnoses and AF classification on physician-assigned diagnoses in medical records corresponding to ICD-10-CM codes for AF in the hospital discharge or outpatient databases. The diagnoses were made based on the medical history, 24-h monitoring, and ECG on admission. We counted the estimated glomerular filtration rate (eGFR) using the CKD-EPI formula, and chronic kidney disease (CKD) was assessed according to the KDIGO 2012 Clinical Practice Guideline for the Evaluation and Management of Chronic Kidney Disease [11]. The diagnosis of metabolic syndrome was based on the current criteria [12]. The diagnosis of other coexisting conditions was made based on medical history, physical examination results, and additional tests by the attending physician; it was not re-examined at the time of inclusion into the study. The medications prescribed at discharge were divided into four groups: acetylsalicylic acid (ASA), dual antiplatelet therapy (DAPT), vitamin K antagonists (VKAs), and direct oral anticoagulants (DOACs). DAPT was defined as taking acetylsalicylic acid (ASA) with a P2Y12 inhibitor (clopidogrel). The group of DOACs comprised dabigatran, rivaroxaban, and apixaban.

The study group was thoroughly characterized clinically, and the CHA_2_DS_2_-VASc, HAS–BLED, and 2MACE scores were calculated at the time of inclusion into the study according to current guidelines [6,7,8].

We checked the usefulness of CHA_2_DS_2_-VASc, HAS–BLED, and 2MACE scores to predict total mortality in the entire study population and in the subgroup of patients with AF.

The study protocol conformed to the ethical guidelines of the Declaration of Helsinki and to the STROBE guidelines [13]. Additionally, it was approved by the local bioethics committee of the Medical University of Białystok (Approval No. R-1-002/18/2019) and registered in the database of clinical studies (www.clinicaltrials.gov (accessed on 1 July 2022); identifier: NCT04541498).

### 2.2. Mortality Data

Data on all-cause mortality were collected from the National Statistical Office in Poland. The records included information on the date and the causes of deaths recorded (codes in the International Classification of Diseases (ICD)—10th Revision).

### 2.3. Statistical Analysis

We used the Kolmogorov–Smirnov test to assess the distribution of variables. Data were presented as medians (Me) and interquartile range (IQR) for not normally distributed continuous variables, and as the number (N) of cases and percentage (%) for categorical variables. The statistical significance of differences between dead and alive patients was determined using the Chi^2^ test and Wilcoxon signed-ranks test.

To compare multiple subgroups for non-normally distributed variables, we applied the Kruskal–Wallis test with multiple pairwise comparisons using the Steel–Dwass–Critchlow–Fligner procedure, whereas for the comparison of categorical variables, the χ^2^ test was used.

The accuracy of the CHA_2_DS_2_-VASc, HAS–BLED, and 2MACE scores in predicting the outcomes was assessed using the area under curve (AUC) receiver operator characteristic curves (ROC). To check the predictive value of modified models, a ROC analysis was also carried out. The results are presented as plots with AUC values.

The Kaplan–Meier statistic was used for graphical assessment of time-dependent mortality according to the CHA_2_DS_2_-VASc and 2MACE scores and the presence of AF. The cut-off point was defined according to Youden’s J statistic [14]. For all analyses, we set the level of statistical significance at *p* < 0.05.

The two-tailed *p*-value < 0.05 was considered statistically significant.

All analyses were performed using XL Stat (Addinsoft, 2020, version 2020.03.01, New York, NY, USA) and MS Excel (Microsoft, 2020, version 16.40, Redmond, WA, USA).

## 3. Results

### 3.1. Study Population, Follow-Up, and Overall Mortality

Out of 26,985 patients admitted for coronary angiography in the study period, after the application of prespecified inclusion (patients who had coronary angiography performed due to the exacerbated angina) and exclusion criteria (acute coronary syndromes, Takotsubo cardiomyopathy, a history of ischemic heart disease, patients referred for coronary angiography before heart valve surgery, and prior cardiosurgical valve replacement), 8288 patients were initially considered for study participation. Additionally, 879 study candidates were excluded. In detail, we excluded foreign patients, residents of other countries or patients who moved to other countries, those with incomplete documentation of contact details, and those without the Polish Universal Electronic System for Registration of the Population. In this population (*n* = 7409), 1244 patients died during the follow-up (16.8%). We also excluded 43 patients with external causes of death (ICD-10: V00–Y98). A total of 7367 patients were included in the final analysis.

More than half of the finally analyzed cohort were men, and the median (IQR) age was 64 years (58–73). Approximately one out of five study participants were diagnosed with AF and CAD, with significant stenosis present in almost 40% of the study population. Detailed characteristics of the analyzed cohort are provided in Table 1.

The median duration of the follow-up was 2029 days (range: 1283–3059 days).

The death rate was 16.3%. The most common cause of death in the study population was chronic ischemic heart disease, followed by lung cancer and ischemic stroke (Table 2).

### 3.2. Comparison between Dead and Alive Study Participants

Comparing alive and dead study participants, we can observe that those in the latter group were older, more often had AF, diabetes, chronic kidney disease (CKD), CAD with significant stenosis, left ventricular systolic dysfunction, and a history of stroke. The patients who died were also more likely to be prescribed VKAs, while therapy with DOACs was numerically more common in this group. A detailed comparison of dead and alive patients is shown in Table 1.

### 3.3. CHA_2_DS_2_-VASc, HAS–BLED, and 2MACE Scores as Predictors of All-Cause Mortality

The value of all CHA_2_DS_2_-VASc, HAS–BLED, and 2MACE scores were significantly higher in patients who died than in alive study participants (Table 1). Additionally, we noticed increased all-cause mortality in the overall study population and AF (+) subgroup with higher values of all compared scores (Figure 1 and Figure 2).

Subsequently, we tested different scores in the overall study population and in the subgroup of AF patients for predicting all-cause mortality.

The ROC curve analysis demonstrated the greatest predictive value for all-cause mortality for the CHA_2_DS_2_-VASc score in the overall study population (area under the curve [AUC] = 0.665; 95% confidence interval [95%CI] 0.649–0.681). In contrast, we observed that the HAS-BLED score had the lowest predictive value for all-cause mortality in the overall study population (AUC = 0.615; 95%CI 0.559–0.631) and in the AF (+) population (AUC = 0.605; 95%CI 0.576–0.635). When considering the overall study population, the 2MACE score was numerically inferior when compared to the CHA_2_DS_2_-VASc score in predicting long-term all-cause mortality (AUC = 0.656, 95%CI 0.619–0.681).

In the case of the overall study population, but not the AF (+) subgroup, our analysis showed statistically significant differences between AUCs for the CHA_2_DS_2_-VASc and 2MACE vs. HAS-BLEED scores (Figure 3 and Figure 4).

The optimal CHA_2_DS_2_-VASc score cut-off point for prediction of all-cause mortality, both in the overall study population and in the AF (+) subgroup, was 3. Based on this threshold, the study participants were divided into two groups, with approximately two-thirds of them falling into the CHA_2_DS_2_-VASc ≥ 3 group. The Kaplan–Meier survival analysis showed a significantly higher all-cause mortality rate in the CHA_2_DS_2_-VASc ≥ 3 group compared to the patients with a lower score (OR = 3.05; 95%CI 2.6–3.6; *p* < 0.001) (Figure 5). Additionally, we established sensitivity (84.1%; 90.3%), specificity (36.6%; 19.5%), positive predictive value (20.6%; 29.6%), and negative predictive value (92.2%; 84.4%) for the prediction of all-cause mortality in study participants with CHA_2_DS_2_-VASc score ≥ 3 (the first and second value in the brackets refer to the overall study population and to the AF (+) subgroup, respectively). Additionally, we performed a similar analysis for the 2MACE scale. The optimal cut-off point for the prediction of all-cause mortality was 1. The Kaplan–Meier survival analysis did not show a significantly higher all-cause mortality rate in the 2MACE ≥ 1 group compared to the patients with a lower score (Figure 6).

## 4. Discussion

According to our knowledge, this study is the first to evaluate the long-term prognostic value of all three scores used in patients with AF for the assessment of the mortality risk in a large cohort, including a subgroup with AF. The main finding of our study suggests that the CHA_2_DS_2_-VASc and 2MACE scores can be moderate predictors of long-term all-cause mortality in patients referred for elective coronary angiography. However, in the subgroup of AF patients, we failed to establish their superiority over the HAS–BLED score.

The CHA_2_DS_2_-VASc, HAS–BLED, and 2MACE scores are designed for risk assessment among patients with AF, and they achieve a high predictive accuracy in their primary purpose [15,16,17]. The guidelines of the European Society of Cardiology recommend the use of the CHA_2_DS_2_-VASc score (chronic heart failure, hypertension, age ≥ 75 years (doubled), diabetes, history of stroke or transient ischemic attack (doubled), vascular disease, age 65–74 years, and female gender) to assess the risk of stroke in patients with AF; however, its usefulness is not limited solely to this [18]. Over the years, it has been extensively tested as a predictor of short-term and long-term mortality in various subpopulations, such as in patients with acute ST-segment elevation myocardial infarction or on hemodialysis [19,20,21,22]. Additionally, its predictive value was compared to other scores directly related to AF.

Since AF is associated with cardiovascular risk factors and was itself recognized as an independent predictor of mortality, many studies have focused on testing these scores only in patients with AF [23,24,25]. Morrone et al. showed that the CHA_2_DS_2_-VASc and HAS-BLED scores had similar predictive values for mortality in patients with AF, and moreover, a combination of those two scores provided the most favorable predictive results [26]. Another recent research on the population with AF demonstrated that the 2MACE score was more likely to predict mortality than CHA_2_DS_2_-VASc [27]. Interestingly, research by Karamchandani et al. showed that the CHA_2_DS_2_-VASc score does not reliably predict in-hospital mortality in patients with new-onset atrial fibrillation [22].

In our study, we compared all three scores in predicting all-cause mortality in the overall study population, as well as in the AF (+) subgroup. In the overall population, the CHA_2_DS_2_-VASc score was numerically superior in predicting long-term mortality, with AUC values for the ROC curve of 0.665. However, when considering the lowest predictive value, we observed different outcomes, as in the overall study population and in the AF (+) subgroup, it was HAS-BLED. In the population of the Białystok Coronary Project, we also clearly showed increased mortality with increased values of all scores. Importantly, the CHA_2_DS_2_-VASc score of 3 or more well identified patients at high risk of death (with OR around 3).

There are studies that go further than testing the scores themselves and additionally look for predictors of mortality, the addition of which could increase the predictive sensitivity of the above-mentioned scores. For example, the addition of renal impairment to the CHA_2_DS_2_-VASc score improved its C-statistic for mortality prediction from 0.70 to 0.72 [28]. In our opinion, the lack of consideration of renal function is an important limitation of many scores, given that CKD is a recognized prognostic factor in cardiovascular disease [29]. The modification of this score had its application even during the global COVID-19 pandemic. A simple change of the gender parameter from female sex to male sex—due to male sex being one of the COVID-19 risk factors—enhanced the scores’ in-hospital mortality predictive value from AUC 0.64 to 0.70 [30].

What is more, when we look at the medications in use among the study group, we can observe that in the cohort, VKAs were more commonly prescribed at discharge, and among patients who died during follow-up, VKA use predominated. The reason for this may be an under-use of DOACs in Poland at the time of study due to their relatively high price and the low economic status of the population. A large study from various European countries from 2011–2016—overlapping with our study period—showed that of the patients taking oral anticoagulants, 67% were on VKAs and only 33% on DOACs [31]. DOACs are characterized by a better safety profile and reduced risk of life-threatening bleeding, especially intracranial bleeding [32]. This is in line with our analysis, as there were fewer deaths in the group of patients with prescribed DOACs.

Our study has some limitations. Firstly, our follow-up includes only data on all-cause mortality; there is no information on other nonfatal clinical events. Secondly, our findings come from a retrospective medium-sized single-center study, including patients referred for elective coronary angiography. Therefore, they should be verified in a prospective multi-center study, including unrestricted individuals, before being applied to the general population. Thirdly, the selection of risk scores in our study was arbitrary. In fact, several other scores have been validated for the prediction of clinical outcomes in AF patients (e.g., ATRIA, CHADS2, R_2_CHADS_2_, HATCH). Fourthly, the follow-up duration in our study was variable and ranged from 1283 to 3059 days (median value 2029 days). Fifthly, in the ROC curve analysis, we did not adjust for the exact timing of death. Sixthly, our analysis performed in the AF (+) subgroup might have been underpowered. Seventhly, the autopsy rate in the study population is unknown. This fact may have some impact on the accuracy of the diagnosis of the underlying cause of death. Eighthly, we were not able to obtain reliable information on the smoking status, diabetes therapy, and changes in pharmacotherapy during follow-up. Lastly, none of these scores take into account external factors, such as air pollution or climate change, which have a large impact on cardiovascular morbidity and mortality [10,33].

## 5. Conclusions

Our study indicates that clinical scores initially validated in AF patients may be useful for predicting mortality in a broader population (e.g., in patients referred for elective coronary angiography). According to our findings, all compared scores have a moderate predictive value. However, in our study, the CHA_2_DS_2_-VASc and 2MACE scores outperformed the HAS–BLED score in terms of the long-term all-cause mortality prediction. Our results should be verified in a prospective multi-center study, including unrestricted individuals, before being applied to the general population.

## Figures and Tables

**Figure 1 ijerph-19-10419-f001:**
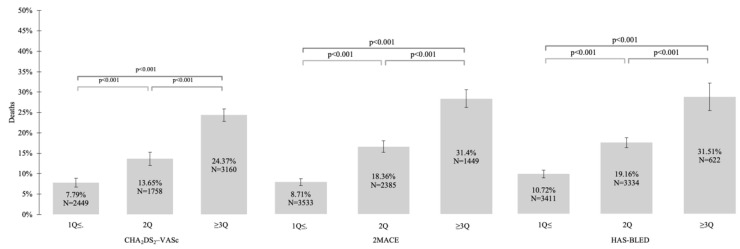
Relationship between all-cause mortality and CHA_2_DS_2_-VASc, 2MACE, and HAS–BLED scores in the overall study population. Abbreviations: Q, quartile.

**Figure 2 ijerph-19-10419-f002:**
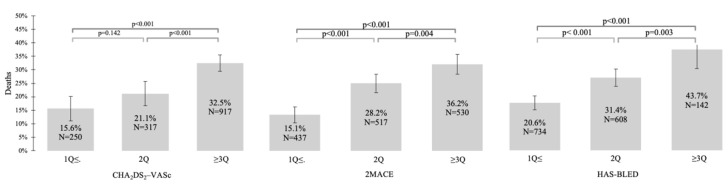
Relationship between all-cause mortality and CHA_2_DS_2_-VASc, 2MACE, and HAS–BLED scores in the group of patients with atrial fibrillation. Abbreviations: Q, quartile.

**Figure 3 ijerph-19-10419-f003:**
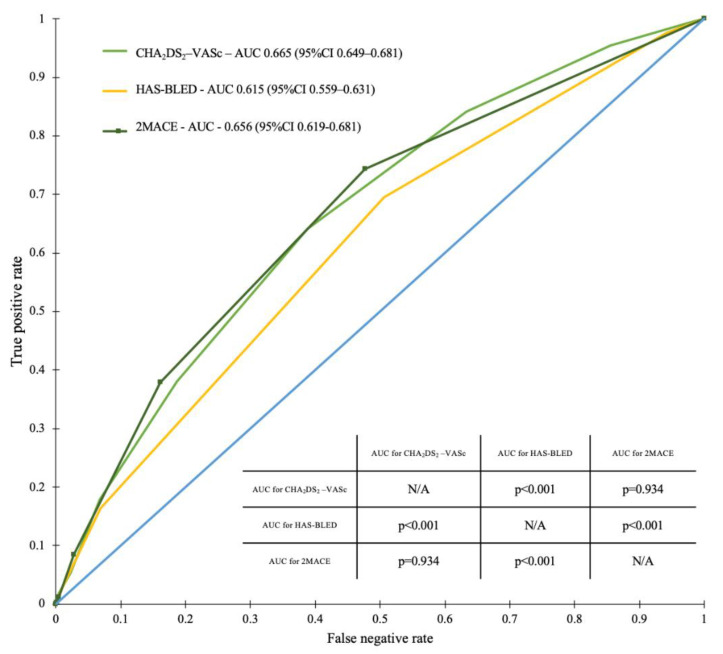
Receiver operating characteristic curves for the CHA_2_DS_2_-VASc, 2MACE, and HAS-BLED scores in predicting all-cause mortality in the overall study population. The inner table shows the *p*-values for comparisons between AUCs for the assessed scores. Abbreviations: AUC, area under the curve; CI, confidence interval; N/A, not applicable.

**Figure 4 ijerph-19-10419-f004:**
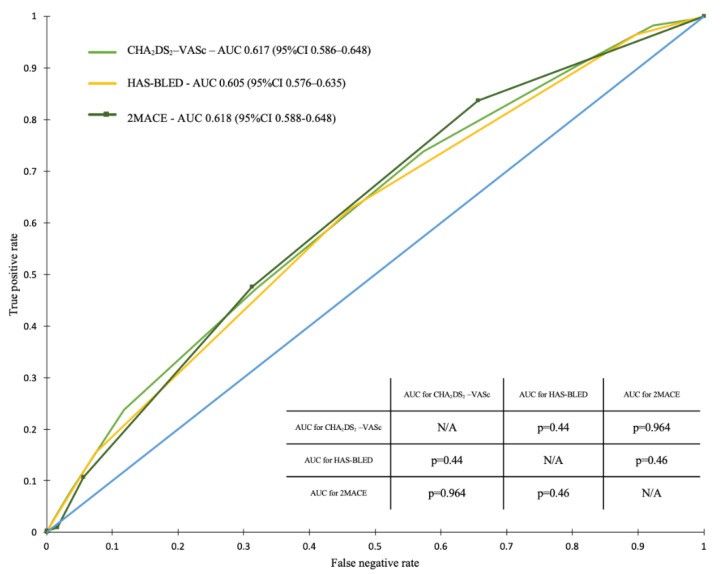
Receiver operating characteristic curves for the CHA_2_DS_2_-VASc, HAS-BLED, and 2MACE scores in predicting total mortality in the group of patients with atrial fibrillation. The inner table shows the *p*-values for comparisons between AUCs for the assessed scores. Abbreviations: AUC, area under the curve; CI, confidence interval; N/A, not applicable.

**Figure 5 ijerph-19-10419-f005:**
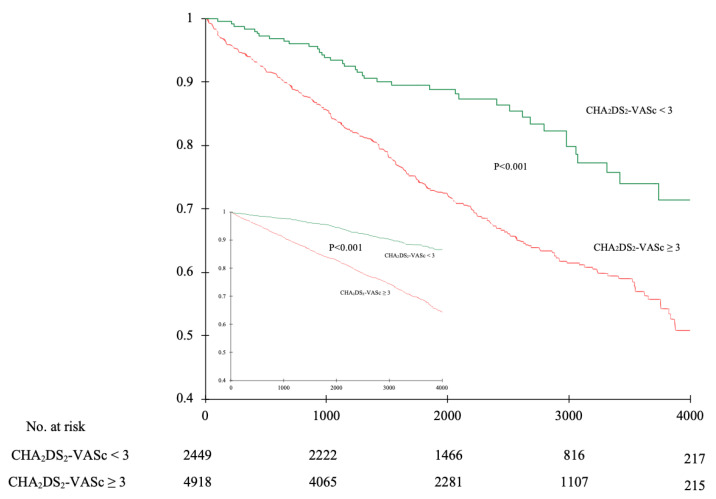
Kaplan–Meier survival analysis of all-cause mortality in relation to the CHA_2_DS_2_-VASc score in the overall study population (large graph). The inner graph represents the comparison in patients with atrial fibrillation.

**Figure 6 ijerph-19-10419-f006:**
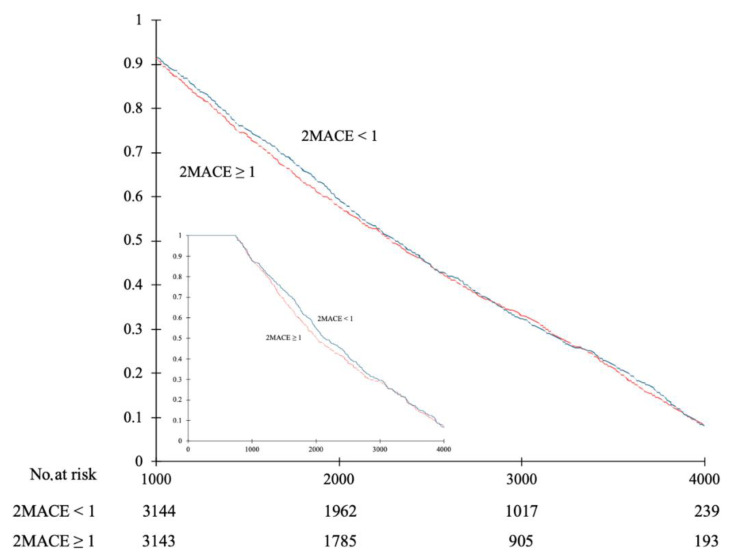
Kaplan–Meier survival analysis of all-cause mortality in relation to the 2MACE score in the overall study population (large graph). The inner graph represents the comparison in patients with atrial fibrillation.

**Table 1 ijerph-19-10419-t001:** Baseline characteristics with a comparison between alive and dead participants of the study.

	Study Participants (*n* = 7367)	Alive Participants (*n* = 6166)	Dead Participants (*n* = 1201)	*p*
Age, years; Me (IQR)	64 (58–73)	64 (57–72)	71 (62–77)	<0.001
Male; % (N)	54 (3978)	51.4 (3168)	67.4 (810)	<0.001
Atrial fibrillation; % (N)	20.1 (1484)	17.5 (1080)	33.6 (404)	<0.001
Hypertension; % (N)	82.8 (6103)	83.3 (5134)	80.7 (969)	0.03
Diabetes mellitus; % (N)	25.5 (1878)	24.2 (1492)	32.1 (386)	<0.001
Hyperlipidemia; % (N)	88.6 (6526)	89.3 (5505)	85.0 (1021)	<0.001
EF < 40%; % (N)	18.3 (1345)	14.3 (884)	38.4 (461)	<0.001
Chronic kidney disease; % (N)	20 (1471)	16.6 (1021)	37.5(450)	<0.001
eGFR, mL/min/1.73 m^2^; Me (IQR)	79 (65–91)	81 (68–92)	71 (54–86)	<0.001
CAD with significant stenosis; % (N)	39.1 (2881)	36.1 (2228)	54.4 (653)	<0.001
ASA prescribed at discharge; % (N)	81.1 (5976)	81.8 (5046)	77.4 (930)	<0.001
DAPT prescribed at discharge; % (N)	21.7 (1599)	21.1 (1300)	24.9 (299)	0.003
DOACs prescribed at discharge; % (N)	4.3 (316)	4.5 (277)	3.3 (39)	0.051
VKAs prescribed at discharge; % (N)	12.4 (914)	10.4 (643)	22.6 (271)	<0.001
Stroke history; % (N)	14.8 (1090)	13.1 (808)	23.5 (282)	<0.001
Bleeding history; % (N)	3.5 (256)	4.0 (249)	6.6 (79)	<0.001
Metabolic syndrome; % (N)	3.6 (272)	3.6 (224)	4.0 (48)	<0.001
CHA_2_DS_2—_VASc score; Me (IQR)	3 (2–4)	3 (2–4)	4 (3–5)	<0.001
HAS-BLED score; Me (IQR)	2 (1–2)	2 (1–2)	2 (1–2)	<0.001
2MACE score; Me (IQR)	1 (0–1)	0 (0–1)	1 (0–2)	<0.001

Abbreviations: ASA, acetylsalicylic acid; CAD, coronary artery disease; DAPT, dual antiplatelet therapy; DOACs, direct oral anticoagulants; EF, ejection fraction; eGFR, estimated glomerular filtration rate; IQR, interquartile range; Me, median; N, number; VKAs, vitamin K antagonists.

**Table 2 ijerph-19-10419-t002:** Most common causes of death in the overall study population (*n* = 7367).

	All Deaths (*n* = 1201)
All, % (N)	16.3 (1201)
Chronic ischemic heart disease, % (N)	22.6 (272)
Lung cancer, % (N)	7.2 (87)
Ischemic stroke, % (N)	5.1 (61)
Instantaneous death, % (N)	4.9 (59)
Myocardial infarction, % (N)	3.9 (50)
Cardiomyopathy, unspecified, % (N)	3.2 (38)
Heart failure, % (N)	3.2 (38)
Hemorrhagic stroke, % (N)	2.3 (28)
Other, % (N)	47.0 (565)

## Data Availability

The data presented in this study are available on request from the corresponding author.

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
