# Peer review of "Comparing Atrial-Fibrillation Validated Rapid Scoring Systems in the Long-Term Mortality Prediction in Patients Referred for Elective Coronary Angiography: A Subanalysis of the Białystok Coronary Project"

_ijerph, 2022, doi:10.3390/ijerph191610419_

Round 1

Reviewer 1 Report

Dear authors,

I think that the paper at hand is well-written and provides interesting information for clinicians. However, a few points should be addressed:

-) Your description of the methods lacks clear in- and exclusion criteria. Even if they are described in the original initial study, a reader will likely not search another paper for information they want to have now.

-) Methods: You describe that the DOAC group consisted of patient receiving dabigatran, apixaban or rivaroxaban; what about edoxaban?

-) Results: What do you mean you excluded patients with "external death"?

-) Table 1, age: Was there really a statistically significant difference? The values are almost identical.

-) Discussion: I feel like that you are trying to convey the message that the scores predict mortality in an overall population; however, your non-AF population is still a highly selected one (CAD and coronary angiography); this should be highlighted.

Author Response

I think that the paper at hand is well-written and provides interesting information for clinicians. However, a few points should be addressed:

We are very grateful for the time and effort invested in reviewing our manuscript as well as the kind words. We have replied to each comment below and edited the manuscript accordingly

  1. Your description of the methods lacks clear in- and exclusion criteria. Even if they are described in the original initial study, a reader will likely not search another paper for information they want to have now.

RESPONSE: We fully agree with this comment. Inclusion and exclusion criteria are now added to the manuscript in the following form:

“Out of 26,985 patients admitted for coronary angiography in the study period, after the application of prespecified inclusion (patients who had coronary angiography performed due to the exacerbated angina) and exclusion criteria (acute coronary syndromes, Takotsubo cardiomyopathy, a history of ischemic heart disease, patients referred for coronary angiography before heart valve surgery, and prior cardiosurgical valve replacement), 8,288 patients were initially considered for study participation.”

  1. Methods: You describe that the DOAC group consisted of patient receiving dabigatran, apixaban or rivaroxaban; what about edoxaban?

RESPONSE: We thank the Reviewer for this question. Edoxaban was omitted by us because it is not available in general use in Poland.

  1. Results: What do you mean you excluded patients with "external death"?

RESPONSE: From the further analysis we excluded patients with external causes of death with ICD-10: V00–Y98. This information is now added to the manuscript in the following form:

“We also excluded 43 patients with external causes of death (ICD-10: V00–Y98).”

  1. Table 1, age: Was there really a statistically significant difference? The values are almost identical.

RESPONSE: We thank the Reviewer for this comment. An error occurred during data transfer and the same value was given for age in the group of patients who died as for the whole study population. The results in the manuscript were checked with the analysis and we have corrected this incorrect value. We also confirm the P-value for age with the corrected value.

  1. Discussion: I feel like that you are trying to convey the message that the scores predict mortality in an overall population; however, your non-AF population is still a highly selected one (CAD and coronary angiography); this should be highlighted.

RESPONSE: We thank the Reviewer for this comment. We believe that we have addressed this message in the first paragraph of the discussion section which emphasized the group to which our findings apply, namely:

“The main finding of our study suggests that CHA2DS2-VASc and 2MACE scores can be moderate predictors of long-term all-cause mortality in patients referred for elective coronary angiography.”

As well as in the ending of the first sentence from the Conclusions:

“Our study indicates that clinical scores initially validated in AF patients may be useful for predicting mortality in a broader population (e.g., in patients referred for elective coronary angiography).”

We are aware of the limitations of this study which we emphasized at the end of the discussion, highlighting the importance of verifying our results in the general population:

“Secondly, our findings come from a retrospective medium size single-center study including patients referred for elective coronary angiography. Therefore, they should be verified in a prospective multicenter study including unrestricted individuals before being applied to the general population.”

Reviewer 2 Report

This study assessed the utility of CHA2DS2-VASc, HAS–BLED, and 2MACE scores in predicting long-term mortality in the population of Białystok Coronary Project, including AF patients. The manuscript requires some modifications before being published. Comments:

  1. As the authors the scores have been developed to assess the risk of: i) thromboembolic complications (CHA2DS2-VASc), ii) major bleeding (HAS-BLED), and iii) composite outcomes (2MACE). Strictly speaking, these scores are not specific for prognosis (death). Why not choose GRACE score or Duke score?

  2. The median duration of the follow-up was 2029 days (range: 1283-3059 days). The follow-up frequency is not clear.

  3. I suspect there were some statistical mistakes in Table 1, such as the age, P<0.0001 (The statistical significance is absent from the data) and Hypertension prevalence. Please confirm the analysis.

  4. The ROCs were black and white, and I suggest use different styles such as solid or dashed line to distinguish them.

  5. The accuracy of CHA2DS2-VASc was close to that of 2MACE from ROC. The Kaplan-Meier survival analysis of 2MACE should also be performed.

Author Response

This study assessed the utility of CHA2DS2-VASc, HAS–BLED, and 2MACE scores in predicting long-term mortality in the population of Białystok Coronary Project, including AF patients. The manuscript requires some modifications before being published. Comments:

We are very grateful for the time and effort invested in reviewing our manuscript as well as the kind words. We have replied to each comment below and edited the manuscript accordingly

  1. As the authors the scores have been developed to assess the risk of: i) thromboembolic complications (CHA2DS2-VASc), ii) major bleeding (HAS-BLED), and iii) composite outcomes (2MACE). Strictly speaking, these scores are not specific for prognosis (death). Why not choose GRACE score or Duke score?

RESPONSE: Our first study from which the study group originated implied that AF was associated with the absence of significant coronary lesions on angiography, reflecting difficulties with qualifying patients with AF for invasive CCS diagnostic workup. With such a specific group of patients, we decided to test the predictive value for mortality of atrial fibrillation-specific scales to assess their usefulness beyond their original purpose.

  1. The median duration of the follow-up was 2029 days (range: 1283-3059 days). The follow-up frequency is not clear.

RESPONSE: We thank the Reviewer for this question. The follow-up range is one of the limitations of our study which is highlighted in the discussion. Poland's Central Statistical Office provides data on individual causes of death for 1.5 years back from the time of the request. Therefore, in the middle of 2021, we obtained data reaching 01.01.2019, hence the frequency of follow-up for our study group. The next mortality assessment is planned for 2023.

  1. I suspect there were some statistical mistakes in Table 1, such as the age, P<0.0001 (The statistical significance is absent from the data) and Hypertension prevalence. Please confirm the analysis.

RESPONSE: We thank the Reviewer for this comment. An error occurred during data transfer and the same value was given for age in the group of patients who died as for the whole study population. The results in the manuscript were checked with the analysis and we have corrected this incorrect value. We also confirm the P-value for age with the corrected value as well as the hypertension prevalence.

  1. The ROCs were black and white, and I suggest use different styles such as solid or dashed line to distinguish them.

RESPONSE: We thank the Reviewer for this suggestion. The ROCs are now in color style for better clarity.

  1. The accuracy of CHA2DS2-VASc was close to that of 2MACE from ROC. The Kaplan-Meier survival analysis of 2MACE should also be performed.

RESPONSE: We thank the Reviewer for this suggestion. According to this, we performed the Kaplan-Meier survival analysis for 2MACE and it is featured in Figure 6. In addition, it is described in the results as follows:

“Additionally, we performed a similar analysis for the 2MACE scale. The optimal cut-off point for the prediction of all-cause mortality was 1. The Kaplan-Meier survival analysis did not show a significantly higher all-cause mortality rate in the 2MACE ≥ 1 group compared to the patients with a lower score.”

Round 2

Reviewer 2 Report

I still have a comment. The optimal cut-off point for the prediction of all-cause mortality was 1 for 2MACE, but the K-M survival analysis is not good. 

How to choose the cut-off point ? Is it from ROC? Please describe it in Statistical Analysis.

Author Response

Yes, we used ROC analysis using statistical software XLSTAT. The cut-off point (Youden) is the maximum sum of Sensitivity and Specificity (for the 2MACE the values were 1.18 for 1 point in the 2MACE scale, 1.16 for 2 points in 2MACE). We performed the K-M curves based on the calculated points. The following publication has been added to the description of statistics in the Materials and Methods section (2.3. Statistical Analysis).

https://www.jstor.org/stable/20486002 

[14] Schisterman, E. F.; Perkins, N. J.; Liu, A.; Bondell, H. Optimal Cut-Point and Its Corresponding Youden Index to Discriminate Individuals Using Pooled Blood Samples. Epidemiology 2005, 16(1), 73–81.